# Use of Motivational Interviewing in Older Patients with Multiple Chronic Conditions and Their Informal Caregivers: A Scoping Review

**DOI:** 10.3390/healthcare11121681

**Published:** 2023-06-07

**Authors:** Beatrice Albanesi, Michela Piredda, Valerio Dimonte, Maria Grazia De Marinis, Maria Matarese

**Affiliations:** 1Department of Public Health and Pediatrics, University of Torino, 10126 Turin, Italy; beatrice.albanesi@unito.it (B.A.); valerio.dimonte@unito.it (V.D.); 2Department of Medicine and Surgery, Campus Bio-Medico University of Rome, 00128 Rome, Italy; m.piredda@unicampus.it (M.P.); m.demarinis@policlinicocampus.it (M.G.D.M.); 3Palliative Care Centre ‘Insieme nella Cura’, Campus Bio-Medico University Hospital Foundation, 00128 Rome, Italy

**Keywords:** motivational interviewing, multiple chronic conditions, older patients, informal caregivers, self-care, scoping review

## Abstract

The use of motivational interviewing is relatively new in multiple chronic conditions (MCCs). A scoping review was conducted according to JBI methodology to identify, map and synthesize existing evidence on the use of motivational interviewing to support self-care behavior changes in older patients with MCCs and to support their informal caregivers in promoting patient self-care changes. Seven databases were searched, from database inception to July 2022, for studies that used motivational interviewing in interventions for older patients with MCCs and their informal caregivers. We identified 12 studies, reported in 15 articles, using qualitative, quantitative, or mixed-method designs, conducted between 2012 and 2022, describing the use of motivational interviewing for patients with MCCs. We could not locate any study regarding its application for informal caregivers. The scoping review showed that the use of motivational interviewing is still limited in MCCs. It was used mainly to improve patient medication adherence. The studies provided scant information about how the method was applied. Future studies should provide more information about the application of motivational interviewing and should address self-care behavior changes relevant to patients and healthcare providers. Informal caregivers should also be targeted in motivational interviewing interventions, as they are essential for the care of older patients with MCCs.

## 1. Introduction

The number of older people suffering from multiple chronic conditions (MCCs), defined as the simultaneous presence of two or more chronic conditions in the same individual [1], is increasing worldwide. In Europe, 65% of people aged 65 years and over [2], and in the USA, 81% of people over 65 years [3] suffer from at least two chronic conditions, and these numbers are expected to increase due to population ageing [4]. Suffering from two or more health chronic conditions is associated with a higher use of healthcare services, greater medication prescriptions and specialized medical visits, with increased costs for healthcare systems [3,5]. Patients affected by chronic conditions need to perform daily self-care behaviors. According to the Middle-Range Theory of Self-Care in Chronic Illness [6], self-care behaviors are those that chronically ill people perform (1) to maintain physical and psychological health stability (e.g., taking medications as prescribed, eating healthily, coping with stress, being physically active, getting enough sleep, avoiding the use of tobacco); (2) to monitor disease symptoms (e.g., checking blood sugar or blood pressure, weighing themselves daily); and (3) to manage symptoms when they appear (e.g., consulting a healthcare provider or taking medication to provide relief [6].

Chronically ill patients, especially when older, are frequently supported in their self-care behaviors by informal caregivers, usually a family member or close friend, who collaborate with patients performing self-care behaviors, or providing the care for them when necessary [7]. The performance of adequate self-care behaviors in older patients with MCCs and the support provided by informal caregivers to patient self-care can be hampered by several factors, such as a lack of knowledge of how to keep chronic diseases stable, monitor signs and symptoms and manage them when they get worse; inability to distinguish between signs and symptoms of different chronic diseases; and lack of motivation to change their behaviors [8,9]. 

Several interventions have been proposed to promote self-care behaviors and improve health outcomes of patients with chronic conditions [10]. Among the interventions aimed at supporting autonomy and engaging people in identifying strategies to deal with their health conditions, motivational interviewing (MI) has been widely used in clinical practice and research [11]. MI is a collaborative, person-centered, goal-oriented style of communication with particular attention to the language of change in the person [12]. When a person presents ambivalence about making health-related behavioral changes, MI helps to reinforce personal motivation and reach specific health goals by eliciting and exploring the personal reasons for changing in an atmosphere of acceptance and compassion [12]. The spirit of MI is based on the following four key interrelated elements: (1) the partnership between patients and counselor, (2) acceptance of what the patients contribute, (3) compassion to actively promote the patients’ welfare and prioritize their needs, and (4) elicitation of the motivations to change that are already present in patients [12]. MI was developed in the 1980s to motivate people to address alcohol addiction [13]. Its clinical application was later successfully extended to a wide range of maladaptive behaviors, such as drug abuse [14], gambling [15], and eating disorders [16]. MI has also been used extensively to promote changes in lifestyle behaviors, such as physical activity [17,18], smoking cessation [19], weight loss [20] and oral hygiene [21]. Moreover, it has been applied to promote changes in the management of chronic conditions, such as pain [22], heart failure [23], chronic obstructive pulmonary disease [24], diabetes [25], and hypertension [26]. MI has also been used for family caregivers to improve child care at home, including interventions aimed at preventing childhood obesity [27], reducing childhood caries [28], treating eating disorders [29] and combatting child vaccination hesitancy [30]. MI interventions have also been directed at informal caregivers to support self-care behavior changes in patients affected by chronic conditions, such as heart failure [31,32,33] and diabetes [34,35].

A few reviews have been carried out to synthesize the evidence on the use of MI in chronic conditions. One systematic review, conducted to assess the effectiveness of MI in promoting lifestyle changes, did not identify any study conducted on older patients with MCCs [36]. Another systematic review evaluated the effect of MI on medication adherence in older patients and identified only a few studies conducted on patients with more than one chronic condition [37]. One scoping review explored the application of technology in MI interventions [38] and another evaluated the quality of randomized control trials (RCT) applying MI [39], but these only included studies carried out on patients affected by a single chronic condition. To our knowledge, no review has described the use of MI to support informal caregivers in promoting self-care behavior changes in older patients with MCCs.

Hence, to date, we do not know exactly to what extent MI has been used in older patients with MCCs, what self-care behaviors have been targeted, how the method has been applied, what aspects have been addressed by participants in the MI sessions, and whether and how informal caregivers of chronically ill older patients have been involved in interventions to promote patients’ self-care behavior changes. As we had a broad research question and we wanted to identify possible gaps in current knowledge, a scoping review was the most appropriate method to investigate the topic. Therefore, the aim of this scoping review was to identify, map and synthesize existing evidence on the use of MI to promote self-care behavior changes in older patients with MCCs and to support their informal caregivers in patient behavior changes. In particular, we wanted (1) to identify the MCCs that MI was used for, (2) the self-care behaviors targeted and the outcomes evaluated, (3) how the MI method was applied (i.e., type of interventions, delivery modes, type of providers, provider training, fidelity assessment), (4) whether and how informal caregivers were involved in MI interventions, and (5) the factors reported by participants that influenced behavioral changes.

## 2. Materials and Methods

The scoping review was conducted in accordance with the methodology for scoping reviews developed by the international organization JBI [40], formerly named the Joanna Briggs Institute, and reported following the Preferred Reporting Items for Systematic Reviews and Meta-Analyses extension for Scoping Reviews (PRISMA-ScR) checklist [41] (Appendix A). The scoping review protocol was registered on the Open Science Framework (OSF) (registration DOI: 10.17605/OSF.IO/CAEJ6).

### 2.1. Eligibility Criteria

We included studies that responded to the following inclusion and exclusion criteria.

*Participants.* We considered studies of patients aged 65 years and over; we also considered studies where more than 80% of the enrolled patients were aged 65 years and over, since studies may not establish a specific age for inclusion criterion, but older patients are generally included in such studies, as MCCs are more frequent in older populations. The older patients could be affected by any combination of chronic conditions. We did not distinguish between comorbidity, which implies the presence of an index disease to which coexisting diseases are related, and multimorbidity, since in clinical practice, patients suffering from more than one chronic condition, whether they have a common etiology, need to perform complex self-care behaviors. As a guide for the identification of chronic conditions to include in the review, we considered the list of chronic conditions developed by the Office of the Assistant Secretary of Health (OASH), as it is a classification system broadly used in international literature [42]. We excluded patients with diagnoses of dementia and psychiatric conditions (i.e., schizophrenia, bipolar syndrome), since these conditions compromise the awareness of health problems, and the reasoning and problem-solving skills needed to participate in an MI intervention. We also excluded people with drugs/substance and alcohol abuse and their caregivers, as MI in these population is directed towards reducing maladaptive behaviors and not towards promoting adaptive health-related behavior changes (i.e., dietary changes, or physical activities level modification) [43], which were the self-care behaviors we were interested in. We excluded studies where patients were affected by a single chronic condition, or not all the patients included in the study were affected by more than one chronic condition, or the number of the chronic diseases was not clearly identifiable. Moreover, we included studies that considered MI interventions directed to informal caregivers of older patients with MCCs, aged 18 and over, either in association with patients or as a specific target of the intervention. We excluded informal caregivers with dementia and psychiatric conditions, and caregivers of patients affected by these conditions, due to the difficulty in conducting MI interventions and evaluating their effects on self-care behavior changes.

*Concept.* We included studies that used MI as a method to motivate patients to change their self-care behaviors by exploring and resolving ambivalence (e.g., understanding the importance of taking medications as prescribed, but not doing so; wanting to increase physical activity, but not finding the time to do it). Changes in self-care behaviors can be self-determined by chronically ill patients to meet their own goals, or they may reflect evidence-based recommendations reached by mutual agreement between patients and healthcare providers [6]. To be included, the intervention had to be named as MI and conducted on the basis of the following four processes described by the original developers: (a) engaging, to create a helpful connection and working relationship; (b) focusing, to develop and maintain a specific direction in the conversation about changes; (c) evoking, to produce personal motivations for change; and (d) planning, to develop a commitment to change and a concrete action plan [12]. The MI could be used as a stand-alone intervention or combined in a multicomponent intervention or program. MI could be integrated in a theoretical framework, such as the Transtheoretical Model of Change in which MI can help patients to move from precontemplation and contemplation to preparation and action stages [44], or it could be used without any reference to a conceptual model. We excluded studies that considered motivation as an outcome of the intervention since in these studies, modifications in motivation level, rather than in self-care behaviors, were evaluated.

*Context.* No context was excluded. Studies could be conducted in any healthcare setting (i.e., hospital, long-term care, and primary care), in social and community services, and in any country and culture.

*Types of evidence source.* The scoping review considered quantitative primary studies, including but not limited to descriptive, cohort, case control, experimental and quasi-experimental studies, and qualitative studies, including but not limited to descriptive, phenomenological, action research studies, and mixed-method studies. We excluded systematic, narrative, or scoping reviews, but, as our scope was to derive information from primary studies in which MI was used, we screened their reference lists to identify additional pertinent articles as recommended in JBI methodology (see next paragraph). We did not consider discussion and theoretical papers, abstracts of proceedings, or clinical trial registration. Authors of abstracts or protocol papers were contacted to obtain information about the publication of their studies.

### 2.2. Search Strategy

A three-step search process was followed according to the JBI method [40]. First, an initial search on PubMed (NLM) and CINAHL (EBSCOhost) databases was undertaken to identify articles related to the topic. The words contained in the titles, abstracts, index terms and keywords of relevant articles were analyzed to develop a full search strategy with the support of an expert librarian. Second, a full search was conducted on the following databases: PubMed (NLM), CINAHL (EBSCOhost), EMBASE (Elsevier), SCOPUS, Web of Science, PsycINFO (EBSCOhost), and Cochrane Library. The search strategy including the relevant terms was drawn up and adapted to each database by an experienced librarian. Third, the reference list of each of the articles included was screened to find additional studies. Moreover, the list of RCT articles (updated June 2022) published on the Motivational Interviewing Network of Trainers (MINT) website (https://motivationalinterviewing.org/, accessed on 17 February 2022) was searched to retrieve further studies. The authors of eligible articles were contacted to identify other papers relevant to the review. Articles published in English, French, Spanish, Portuguese and Italian, languages well-known by the research team, from database inception to July 2022, were included. The full search strategy for all databases is provided in Appendix A.

### 2.3. Source of Evidence Selection

After completing the search, the records identified were uploaded into EndNote version 22, and duplicates were removed. The results were screened by reading titles and abstracts. Papers considered relevant were read in full and assessed against the inclusion and exclusion criteria. The web app Rayyan [45] was used to support the screening and selection process, which was reported in a Preferred Reporting Items for Systematic Reviews and Meta-Analyses (PRISMA) flow diagram [46]. Two reviewers conducted the screening and selection processes independently and any disagreement was resolved through discussion or by consulting a third reviewer.

### 2.4. Data Extraction and Analysis

Two reviewers, using data extraction tools developed by the reviewer team, extracted data from the included articles independently. For the quantitative studies, the following data were included: authors’ names and publication year, study country, aim, design, setting, samples, data collection method, targeted health behavior and outcomes, study results, mode of MI delivery, types of providers, provider training process, training, and method fidelity assessment. For the qualitative studies, the following data were included: authors’ names and publication year, country, aim, design, setting, study samples, data collection method and analysis, and findings. Any disagreement arising between reviewers was resolved through discussion or consultation with a third reviewer. Authors of the papers included were contacted to request missing or additional data when needed. The data were charted and reported in tables and graphical form. A narrative summary accompanied the tabulated and/or charted results and illustrated them according to the review’s aims.

## 3. Results

### 3.1. Search Results

The electronic searches yielded 1023 records after removing duplicates. Of these, 997 records were not pertinent and were excluded. Full texts were retrieved for 26 records. Of these, 13 papers were excluded because they did not fulfill the inclusion criteria (see Appendix A for the list of excluded studies). From an additional search on websites and reference lists, 10 further records were identified; of these, 8 were excluded. The main reasons for exclusion were a lower age-range, patients with a single chronic condition, and no MI interventions. Fifteen articles reporting twelve studies were finally found eligible for inclusion in the review (Figure 1). Two articles presented the results of a study conducted on the same sample describing two different outcomes [47,48], and two qualitative articles [49,50] presented an analysis of the contents of the MI sessions of two quantitative studies [51,52].

### 3.2. Study Characteristics

The study characteristics are summarized in Table 1 and Table 2. The articles were published between 2012 and 2022. Quantitative designs were used in 11 articles, qualitative designs in 3 and a mixed method in 1. For the mixed-method study [53], the qualitative and quantitative components of the study were analyzed separately. In the three qualitative articles and in the qualitative component of the mixed method article, cross-sectional designs [49,50,53], and a longitudinal design [54] were used. Regarding the articles reporting quantitative studies, four used a quasi-experimental design [53,55,56,57], and eight a RCT design [47,48,51,52,58,59,60,61] (Table 1). The qualitative data were collected by using individual interviews [49,50,54] and focus groups [60], and analyzed through thematic [60], content [54] and theoretical framework [49,50] analyses. The studies were conducted in Australia [49,50,51,52,60], the USA [53,56,59], Spain [47,48,55], Brazil [58], Italy [61], Turkey [57] and Sweden [54] (Table 2).

We could not identify any study in which MI was used with informal caregivers of older patients with MCCs. One study, describing a patient-centered prescription program for patients with multimorbidity, reported that informal caregivers were included in the program, but did not specify how they were involved in the MI intervention [55]. Therefore, our scoping review reports only the results related to older patients with MCCs.

### 3.3. Characteristics of Older Patients and Multiple Chronic Conditions

A total of 2169 patients participated in the quantitative studies and 99 in qualitative studies. The patients were mainly female, with a mean age ranging from 66 [58] to 83 years [55]. One study specifically targeted older women [60]. In two articles, culturally and linguistically diverse older patients living in Australia [50,51] were recruited, and in three articles, Black and/or African American patients [56,58,60] were included (Table 1 and Table 2). The patients were mainly affected by diabetes mellitus and diseases frequently associated with diabetes, such as chronic kidney disease, cardiovascular diseases (e.g., hypertension, chronic heart failure) (Figure 2). Three articles [49,52,56] considered patients with multimorbidity, and two older patients with polypharmacy [47,48], in some cases without specifying the chronic diseases that affected the patients (Table 1 and Table 2).

### 3.4. Targeted Self-Care Behaviors and Outcomes

In eight articles, the MI was used to promote changes in adherence to medications in low-adherent patients with MCCs [51,51,55,57] with polypharmacy prescription [47,48], and with specific drug prescriptions, such as hypertensive and antidiabetics [56], angiotensin-converting enzyme inhibitors and angiotensin II receptor blockers [59]. In three articles, the MI was aimed to improve changes in disease self-management [53,58,61] and in one to promote physical activity [60] (Table 1 and Table 2).

In the quantitative studies, the primary outcomes measured were as follows: medication adherence [48,51,52,55,57,59], self-reported physical and mental health, physical activity [60], blood pressure [52,56], glycated hemoglobin [56,58], beliefs about medications, quality of life, and health awareness [57], medication errors [47], self-rated health [53], emergency room visits and hospital admissions [61], and medication self-efficacy [51].

### 3.5. Characteristics of the Motivational Interviewing Interventions

*Type of intervention.* The MI was used as a stand-alone intervention in five articles [47,48,54,58,59], and in the remaining as a component of multifactorial interventions (i.e., medication self-management intervention [52]), programs (i.e., patient-centered prescription model [55], medication self-management support program [56], chronic disease self-management program [53], care management program [61], or services (i.e., community pharmaceutical care services [57]. A few studies reported the conceptual framework that the intervention or program was based on, such as the Health Belief Model [51,52], the Behavioral Change Wheel [57], and Cognitive Behavioral Therapy [53].

*MI providers.* Different healthcare providers delivered the MI intervention, including nurses [47,48,49,50,51,52,58,60,61], physicians [47,48], pharmacists [55,57], pharmacist students [59], primary care providers [56], psychologists [53], professional counselors [60], and social workers [54].

*Training.* Six articles [47,48,54,57,58,59] provided information about the type of training, which lasted from 4 hours [57] to 5 weeks [54]; the training was conducted in person except in one study in which it was conducted online [57]. The training was performed through a course/workshop format by members of MINT in two studies [54,59], and in the remaining by experts in MI. Five articles [47,48,57,58,59] provided details regarding the educational methodology, which entailed lectures and experiential exercises, such as a video, demonstration, interactive case presentation, role-playing, and simulation using a trained standardized patient.

*Training effectiveness assessment.* Four articles [47,48,58,59] reported some form of trainee assessment regarding the acquisition of MI competency that was performed at the end of the training program, using standardized instruments [47,48,58].

*MI delivery mode.* The MI interventions were delivered individually face-to-face [47,48,56,56], remotely by phone [49,50,51,52,53,59], or using a combination of both modes [54,57,60,61]. In one study, the delivery mode was not specified [55].

*Exposure time.* The motivational interviewing sessions lasted between 2 [49,52] and 90 min [54], and the frequency of sessions ranged from 2 [56] to 9 [60] in a period of 3 to 6 months. The phone sessions presented the shorter contact time [49,52,59].

*Treatment fidelity assessment.* To verify the extent to which the MI method is implemented as intended, a treatment fidelity measure can be performed, usually by sampling the encounters and measuring the interviewer’s verbal behaviors using structured instrument [62]. No study reported fidelity assessment performed during or at the end of the MI sessions. A few studies reported that the providers had received regular supervision and coaching during the provision of the interventions [47,48,58,59].

A full description of the characteristics of the MI interventions is reported in Appendix A.

### 3.6. Factors Influencing Motivation to Change Self-Care Behaviors

Four articles [49,50,53,54] analyzed the contents of the MI sessions providing information about the issues that were addressed by participants during the encounters with the MI providers (Table 2). Older patients identified personal, clinical, psychological, social, relational, economic, and healthcare system-related factors that could either promote or hinder their self-care behavior changes. These are summarized in Table 3.

*Healthcare system-related factors.* In Brandberg and colleagues’ study [54], the two most important challenges reported by older patients were navigating healthcare systems that are not organized to meet the needs of patients with MCCs and handling the burden of living with multiple illnesses. Patients reported that they were left alone in managing continuity of care after hospital discharge. They dealt with several healthcare professionals with different opinions regarding treatment strategies, and even their general practitioner lacked knowledge of their clinical conditions. Having more healthcare providers in charge of their health increased the fragmentation of care and led to poor information about diagnoses and treatments, medication regimen and side effects. The medical record systems differed between healthcare services, forcing older patients to take on the responsibility of transmitting information on their care to healthcare providers. Moreover, the pharmaceutical policy of offering the cheapest products complicated drug management, as patients struggled to remember the generic names of drugs. During the MI sessions, several strategies on how patients could prepare for encounters with healthcare providers and overcome the lack of care continuity were addressed.

*Personal factors.* During the MI sessions in Williams and Manias’ study [49], older patients reported several personal factors that influenced their motivation and self-confidence in taking the prescribed medicines. Perceptions of their health status, the severity of their diseases, and potentially life-threatening conditions predisposed patients to take their medication, although in some cases, they made personal adjustments [49]. For example, the desire to be in control of their health led patients to modify their prescriptions or use alternative or complementary medicines without sharing their decision with physicians. Prior experience with medicines, knowledge, positive feedback, and optimistic views regarding medicine regimens influenced their motivation to adhere to the prescribed treatment [49].

*Clinical factors.* Older patients reported that fatigue impaired their ability to perform activities of daily living, maintain a social life, and manage the new medical regimen in the first post-discharge weeks [54]. Reduced levels of attentiveness due to a lack of energy and strength caused difficulties in discerning the disease from which their symptoms derived, in some cases delaying their referral to healthcare providers and causing re-hospitalization [54]. In the weeks following discharge, MI sessions were characterized by issues regarding how to manage stress and anxiety caused by feeling a burden to family members, and uncertainty about which healthcare service was responsible for their post-discharge care [54]. Moreover, patients reported that complex treatments and co-occurrence of acute illnesses hampered their treatment adherence, and that underestimating or denying the danger of their symptoms led to reducing or suspending the treatment [49]. Older patients also reported the occurrence of side effects as another factor that often led them to stop taking medications without consulting their physicians [50]. Patients reported several strategies they used to remember to take their medications, such as using a dosette box, illustrations on medication packaging where they were unable to understand English, or maintaining a routine (e.g., taking their medications at meals, or keeping medicines in the same place) [49]. They also asked family members living in the same household for help preparing medications or remembering to take them, and to accompany them to appointments with physicians and to assist with translation. The hope of not becoming more seriously sick was a strong motivation for taking medications. Non-English-speaking participants reported several factors that hampered their medication adherence, for example, the use of dosette boxes and specialist letters written in English reduced their possibilities of knowing more about the prescribed medication [50]. In the work of Halloway et al. [60], older women identified joint pain and acute illnesses, such as colds and influenza, as barriers to being physically active, and as motivation for using a monitoring device that allowed them to track their steps and evaluate the achievement of their goals. They reported the integration of physical activity into sedentary behaviors (e.g., walking on the spot during television commercials, and walking to some places instead of driving) as helpful strategies for increasing physical activity. They identified several benefits of increasing physical activity, such as better overall health, greater opportunities for socializing, and greater confidence in their physical functions (e.g., ability to walk upstairs or to play with their grandchildren) [53].

*Psychological factors.* Attitudes towards medications influenced medication self-efficacy in some older patients [50]. Although patients recognized the importance of taking medications as prescribed, they prioritized medicines according to their perceived importance and to the daily life restrictions that they caused. They felt discouraged by having to take medications for the rest of their lives [50].

*Social factors.* Patients reported belonging to a different culture, family problems, and lack of resources as social factors that interfered with medication self-management [49].

*Interpersonal factors.* In the work of Williams and Manias [49], older patients reported that motivation to take medicines was affected by the all-important relationship with healthcare providers, together with the difficulty of navigating the healthcare system and lack of care continuity.

*Economic factors.* The cost of medications was another factor that impeded adherence to medication regimens, especially when older patients did not have access to state benefits [49,50].

## 4. Discussion

This scoping review aimed to identify to what extent and how MI interventions have been used in research to promote self-care behavior changes in older patients with MCCs and to support their informal caregivers in promoting patient self-care changes. We identified 12 studies, reported in 15 articles, that described the use of MI for patients with MCCs. We could not locate any study that targeted informal caregivers of older patients with MCCs, although one study reported the involvement of caregivers in a self-management program that included an MI intervention but without providing any details regarding their participation. The identified studies were all published in the last 10 years, showing that the use of MI in the context of MCCs is quite recent, although its application in healthcare began more than 40 years ago [63]. This demonstrates a current shift in researchers’ attention from patients with single chronic conditions to those with MCCs, as they represent the most prevalent older population in clinical practice and their number is expected to increase in the coming years [4]. Most of the identified studies considered patients with diabetes and their comorbidities, consistently with the epidemiological data that show the high prevalence of these conditions in older populations worldwide [64]. Our review has demonstrated that MI has been used to enhance personal motivation to change in old patients, as the identified studies included people aged 80 and over, and older patients from different cultural and ethnic backgrounds, confirming that this method can be used at any age [65] and in people from any cultural background [66].

Although MI can be used to promote changes across a wide range of health-related behaviors, the behavior addressed in the identified studies was predominantly adherence to and management of medications. The fact that most of the identified studies addressed medications adherence demonstrates that the healthcare professionals consider it an essential component of self-care [6], since suboptimal medication use can have negative consequences on patient health [55]. In the older patients, the level of non-adherence is high, ranging from 40% to 86%, and several factors may contribute to it, such as patients’ visual, functional, and cognitive impairment, fear of side effects, or economic burden [59]. Healthcare providers need to work collaboratively with patients to negotiate adherence to therapies, based on the best evidence and keeping in mind the adverse effects of medication that can occur in old age. In the qualitative studies included in our review, older participants, especially in cases of polypharmacy and complex regimens, expressed ambivalence about medication use. On the one hand, they knew that medicines were important to prevent exacerbations, reduce symptoms and prevent long-term complications; on the other, personal beliefs and attitudes towards medications, lack of trust in physicians, social and economic issues hampered their adherence. For example, they struggled to accept having to take medications for the rest of their life and they preferred to find alternative solutions without consulting their physicians. The ambivalence of patients towards prescribed treatments is seldom addressed by physicians who employ an authoritative style with older patients. Such physicians assume that the patients will simply follow their prescriptions as they are for their good, rather than actively involve them in treatment choices or share decisions with them [55]. We found only one study that used MI to promote the modification of a lifestyle behavior, that is physical activity in older women with MCCs. Several self-care behaviors are of great importance for patients with MCCs as they can contribute to preventing and reducing the effects of chronic conditions on their daily life. Some of these are as follows: following a healthy diet, smoking cessation, sleeping well, receiving vaccinations and regular checkups, maintaining social connectedness, managing stress, and being able to monitor clinical manifestations of health conditions and to distinguish between different disease symptoms [6]. Studies should be conducted to investigate the effects of MI on a broader spectrum of changes in self-care behaviors that are important for older patients and should not only prioritize those considered important by healthcare professionals, such as medication adherence.

Analysis of the content of MI sessions derived from the qualitative studies provided important information on the factors that can influence motivation in self-care behavior changes in patients with MCCs. For example, chronically ill patients reported that behavior changes were hampered by the healthcare system organization, which do not ensure the continuity of care. Since patients struggle to deal with their chronic diseases all together, the MI can help them identify those health issues that are most important for them, informed by the clinician’s knowledge. In fact, patients and clinicians might have different views regarding the priorities in the management of MCCs. For example, healthcare professionals can prioritize blood pressure control to prevent complications in older patients with diabetes and hypertension, while patients might prioritize the management of chronic pain caused by arthritis, as this interferes with taking care of their grandchildren. In that sense, healthcare professionals should be trained to use MI in their practice to identify the reasons for patient resistance and promote the integration of self-care behaviors in their lives.

Our review showed that MI was used as a single intervention or combined with other interventions or programs. The use of MI in multicomponent interventions is quite common, as MI is a method that can be used in all situations in which people express ambivalence towards changing their behaviors, and in conjunction with all theoretical frameworks based on behavior changes [12]. In the four RCTs that used MI as a stand-alone intervention, authors reported that MI was effective in improving medication adherence and self-management, as found in studies conducted on patients affected by a single chronic condition, such as heart failure and diabetes [23,25]. In the studies combining multiple interventions, due to the complex nature of interventions, it was difficult to determine which program features were responsible for the positive outcomes and, therefore, the direct contribution of MI in determining changes.

In most of the identified studies, we found a limited description regarding how the MI intervention was conducted (i.e., length of intervention, number of sessions, exposure time), especially when MI was integrated into another program, making it difficult to ascertain whether the results could be attributed to MI. A complete description of the intervention should be provided by researchers to allow the accuracy of the MI implementation method to be evaluated by adhering, for example, to international standards of intervention reporting, such as the Template for Intervention Description and Replication (TIDieR) checklist and guide [67]. Moreover, no treatment fidelity assessment was reported in any identified study, although such evaluation is important to assure that the MI is delivered consistently with the method. A few instruments have been developed by researchers to measure interviewers’ adherence to the MI principles such as the Motivational Interviewing Treatment Integrity (MITI) scale that has demonstrated good reliability and validity [68]. In our review, variability was also found regarding the training process of MI providers, with training lengths ranging from four hours to five weeks, and the training methodology. Miller and colleagues [69] highlighted that to be effective, MI training should be long in duration and integrate observation, feedback, and coaching. Moreover, continued practice, supervision and monitoring are recommended in addition to the initial training, to attain and maintain method standards [69]. In the identified studies, MI was prevalently administered face-to-face or via phone. In recent years, other delivery approaches have been suggested for MI, including video calls or video conferences, which proved successful in enhancing patient outcomes [70]. These modes can present some advantages over phone or in-person delivery. On the phone, patients can be more distracted and less involved in conversation; by contrast, during video calls, their attention can be recalled by interviewers and the conversation can be redirected onto behavior changes; moreover, interviewers could also register nonverbal communication [70]. Further research should be conducted to verify the acceptability and effectiveness of MI intervention using these new technological means in older patients with MCCs.

We could not find any study that used MI for informal caregivers to promote changes in the self-care behaviors of patients with MCCs. Most of the research on the use of MI for informal caregivers was conducted on caregivers of patients affected by a single chronic condition, such as heart failure [31,32,33] and diabetes [34,35]. These studies suggest that when informal caregivers are involved in MI interventions, patients are more likely to modify their behaviors and patient health outcomes improve. The involvement of informal caregivers of chronically ill patients as co-care providers is broadly recommended in programs based on the person-centered care approach [71]. Informal caregivers can contribute to chronically ill patient self-care by recommending self-care behaviors (e.g., doing regular exercise, taking the prescribed medicines), or by partially or totally substituting patients when they are unable to perform them alone, for example when they are not autonomous in the activities of daily living or are cognitively impaired (e.g., preparing low-salt foods, measuring glycemia, or calling physicians in cases of exacerbation [72]). Our review highlights the lack of research on the use of MI for informal caregivers and the need to conduct studies to explore the possible effects of involving informal caregivers to improve care of older patients with MCCs.

### Limitations

This scoping review has some limitations. First, we used broader population inclusion criteria than those initially planned in our first preliminary literature search, since we found that studies’ authors did not always report full information about sample age, or they used other criteria to set the age for the older patients. Therefore, enlarging the age inclusion criteria allowed us to include more articles. Second, we included studies in which the MI approach was integrated in a self-management program and the MI was not fully described, limiting the quantity of the information that we could extract; consequently, a few of the included studies provided limited knowledge of how MI intervention and the training process were conducted, and how fidelity to the method was assured. Third, we acknowledge that the number of chronic conditions does not necessarily increase the complexity of self-care behaviors, as there are some chronic conditions that have a low impact on patients’ daily lives and functioning. We decided to include all types of chronic conditions because of the limited literature on MI in MCCs. Fourth, we are aware from the number of RTCs and personal contacts with researchers that there are several ongoing studies on MI in MCCs; the publication of these studies will extend the knowledge of MI in MCCs.

## 5. Conclusions

As an increase in the older patients affected by MCCs is expected over the coming years, interventions to improve self-care behaviors should be developed and tested on this population. Although MI has proved to be effective in favoring behavioral changes across several health conditions, its application in patients with MCCs is limited and its effectiveness has not yet been evaluated. We found that the studies conducted on this population so far reported scarce information about how MI intervention was delivered and how fidelity to the method was guaranteed, particularly when MI was used in multicomponent interventions. Traditional strategies of MI delivery were used in the identified studies; technology-based delivery strategies should be tested to favor the participation of older patients in MI interventions, especially in cases of mobility impairment or attention deficit. Although in the identified studies MI was used mainly to facilitate changes in patient medication adherence, several other self-care behaviors should be targeted by MI, since they are important in helping to keep chronic diseases stable, and in monitoring and managing symptoms in individuals with MCCs. As we did not find any study conducted for informal caregivers, future research should also test MI interventions directed at caregivers, as they are essential in supporting older patients with MCCs to care for themselves and could help motivate patients to change their self-care behaviors.

## Figures and Tables

**Figure 1 healthcare-11-01681-f001:**
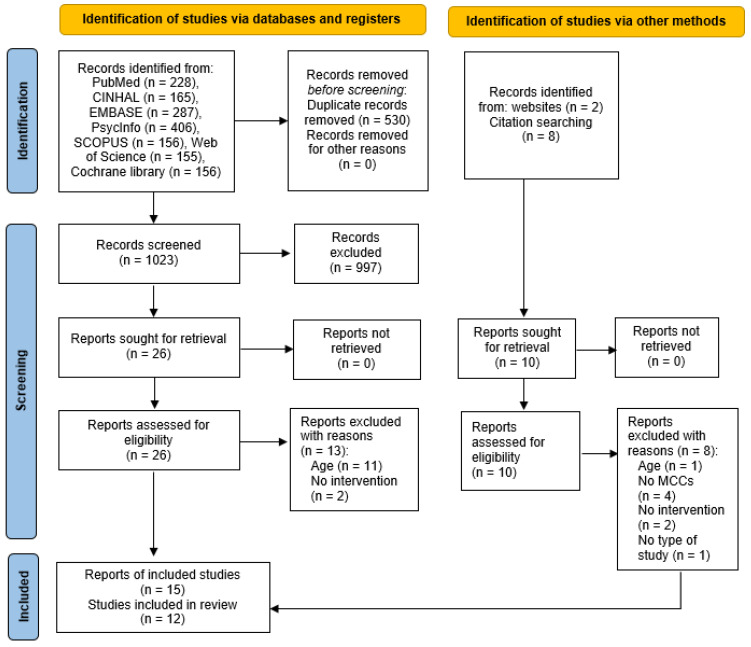
PRISMA Flow Chart.

**Figure 2 healthcare-11-01681-f002:**
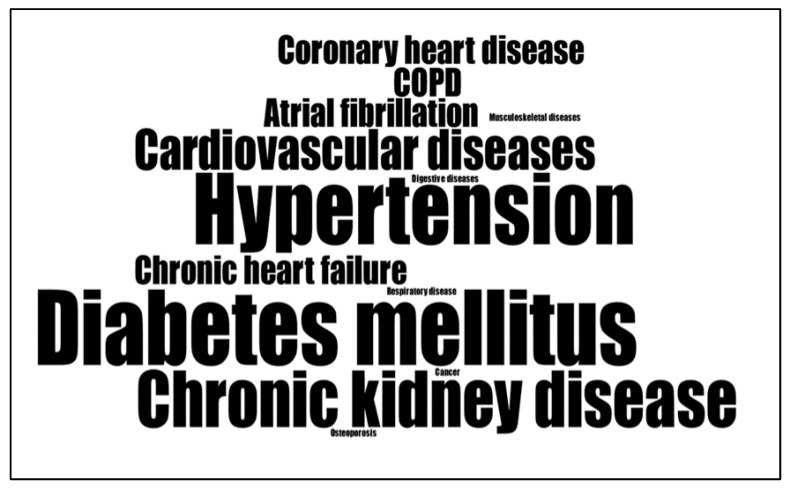
Word cloud describing frequency of chronic diseases considered in the studies.

**Table 1 healthcare-11-01681-t001:** Characteristics of the quantitative articles included in the scoping review.

Reference/ Country	Design	Setting	Aim	Participants	Chronic Conditions a. Number; b. Type	Self-Care Behavior	Outcomes	Intervention/Follow up	Results
Abughosh et al. [59], 2017 USA	RCT	Primary care	To examine the effect of MI conducted by phone by pharmacy students in improving adherence to ACEIs and/or ARBs among Medicare advantage plan patients with DM and hypertension.	743 patients (EG: 248; CG:495) Mean age (SD): 69.79 (10.23) M: 43.74%	a. ns; b. DM and hypertension	Medication adherence	Adherence to ACEI and ARB	MI 6 months	Patients receiving two or more MI phone calls had significantly better medication adherence and less discontinuation in the following 6 months compared with those who did not receive MI phone calls.
González-Bueno et al. [55], 2022 Spain	Quasi experimental before–after study	Intermediate care	To assess the impact of the patient-centered prescription model in medication adherence and effective prescribing in patients with multimorbidity.	93 patients Mean age (SD): 83 (6.1) M: 34%	a. Median: 7 (IQR 4–12); b. ns	Medication adherence	Primary: medication adherence. Secondary: n. medications, regimen complexity, drug burden, patients with ≥2 inappropriate prescribing	Patient-centered prescription model including MI 6 months	The patient-centered prescription model improved medication adherence and effective prescribing in non-institutionalized older patients with multimorbidity and polypharmacy in patients discharged from a rehabilitation center.
Halloway et al. [60], 2020 USA	Mixed method Quasi-experimental pre-post test study	Outpatient clinic	To examine feasibility, acceptability, and preliminary effects of a 24-week lifestyle physical activity intervention for older women.	18 women Mean age (SD): 73.9 (5.3) African American: (61.1%) Non-Hispanic white: (38.9%)	a. Mean (SD): 3.2 (0.9); b. Coronary heart disease, hypertension, atrial fibrillation	Physical activity	Primary: self-reported physical and mental health Physical activity Cardiorespiratory fitness	Intervention including lifestyle physical activity prescription, group meetings, and individual MI phone calls. 24 weeks	Meeting attendance was >72% and retention 94%. Participants rated the program with high satisfaction. There were significant improvements at 24 weeks in self-rated physical health, daily steps, and estimated cardiorespiratory fitness.
Kim et al. [56], 2020 USA	Quasi experimental pre-post test study	Internal medicine residency primary care clinic	To evaluate the impact of an interprofessional medication self-management support program on blood pressure and HbA1c in underserved older patients with uncontrolled hypertension and DM.	50 indigent patients Mean age (SD): 67 (5) M: 22% Black/African American: 88%	a. Mean (DS): 8.26 (2.67); b. DM and hypertension	Medication self-management	Primary: Systolic and diastolic blood pressure and HbA1c level Secondary: adverse effects, n. chronic medications	Medication self-management support program including MI and setting goals 12 months	Significant improvements in systolic blood pressure and HbA1c were observed and sustained following implementation of the medication self-management support program.
Moral et al. [48], 2015 Spain	RCT	Health care setting and home	To evaluate the effectiveness of MI in improving medication adherence in over 65 patients treated by polypharmacy.	154 patients (EG: 70, CG: 84) Mean age (SD): 76 (5.9) M: 31.2%	a. Mean (SD): 5 (2.59; b. ns	Medication adherence	Primary: medication adherence	MI 6 months	MI was more effective than traditional intervention (advice and information) in improving medication adherence.
Okuyan et al. [57], 2021 Turkey	Quasi-experimental study	Primary care	To evaluate the impact of theory-based, structured, standardized pharmaceutical care services led by community pharmacists on patient-related outcomes in older Turkish adults.	52 patients Mean age (SD): 73.4 (5.4) M: 51.9%	a. Mean: 3.7 b. COPD, hypertension, type 2 DM, osteoporosis	Medication adherence	Primary: medication adherence, beliefs about medications, quality of life, health awareness. Secondary: changes in lifestyle behaviors, immunization, inappropriate prescribing and satisfaction with services.	Community pharmacist-led pharmaceutical care services including community pharmacists-led medicine bag check-up, patient medicine card, patient education and counseling services including MI 3 months	The pharmaceutical care services significantly improved medication adherence, beliefs about medication, and quality of life in older adults. The services also decreased potential inappropriate prescribing, frequency of falls, and hospitalization at 3 months and increased the rate of seasonal influenza and pneumococcal vaccines in older adults.
Pérula de Torres et al. [47], 2014 Spain	RCT	Healthcare setting and home	To evaluate the effectiveness of MI intervention to reduce medication errors in over 65 chronically ill patients with polypharmacy.	154 patients (EG: 70, CG: 84) Mean age (SD): 76 (5.9). M: 31.2%	a. Mean (SD): 5 (2.5); b. ns	Medication errors	Primary: medication errors	MI 6 months	MI was more effective than usual approach in reducing medication errors in over 65 patients with polypharmacy.
Reed et al. [53], 2018 Australia	Parallel-group RCT	Primary care	To determine whether a clinician-led chronic disease self-management support program improves the overall self-rated health level of older Australians with multiple chronic health conditions.	254 patients (EG: 127, CG: 127) Mean age: ns Age range: 60 ≥ 85 M: 24% Australian: 76% European: 24%	a. Mean (SD): 4.4 (0.12); b. CVD, respiratory, musculoskeletal, psychological, digestive, kidney diseases, type 1 or 2 DM, cancer	Self-management	Primary: self-rated health Secondary: health status (fatigue, pain, health distress, energy, depression, illness intrusiveness), health behaviors (exercise, medication adherence), self-efficacy, health education, and health care utilization	Chronic disease self-management support program based on cognitive behavioral therapy and MI 6 months	The program improved the self-reported health of older patients. No improvement in other health outcomes was found.
Steffen et al. [58], 2021 Brazil	RCT	Primary care Health units	To evaluate the effectiveness of MI in individual nursing consultations for management of type 2 DM with hypertension	189 patients (EG: 101, CG: 88) Mean age (SD): 66 (9.91) M: 38.4% Black: 13.5%	a. ns; b. type 2 DM and hypertension	Self-management	Primary: HbA1c Secondary: blood pressure, treatment adherence	MI 6 months	MI was more effective than usual care in improving blood pressure and treatment adherence levels (correct intake of prescribed drugs, healthy diet, physical activity), regular follow-up and personal involvement. MI was as effective as usual care in reducing HbA1c levels.
Tiozzo et al. [61], 2019 Italy	RCT	Primary care	To describe the impact of a care management program in reducing emergency room and hospital admissions among patients affected by HF and multimorbidity.	488 patients (EG: 244, CG: 244) Mean age (SD): 78 (9.17) M: 48%	a. mean (SD): 4.3 (2.36); b. HF and other chronic conditions	Self-management	Primary: emergency room visits and hospital admissions	Care management program including MI 1 year	The care management program reduced emergency room visits and hospital admissions in older patients with multimorbidity.
William et al. [52], 2012 Australia	RCT	Primary care	To test feasibility and impact of a multifactorial medication self-management intervention to improve blood pressure control and medication adherence in adults with DM and chronic kidney disease.	80 patients (EG: 39, CG: 41) Mean age (SD): 67 (9.6) M: 56.3% Non-Australians: 63.8%	a. Mean (SD): 7.9 (2.5); b. type 1 or 2 DM and chronic kidney disease, or diabetes kidney disease and hypertension	Medication self-management	Primary: Medication adherence and blood pressure control Secondary: attrition rates, intervention participation, satisfaction with the intervention	Medication self-management intervention including self-monitoring of blood pressure, individualized medication review, 20 min DVD, and MI phone contact 3 months	The intervention was acceptable to participants, and feasible. There were no statistically significant differences between groups in reduction in blood pressure level and medication adherence.
William et al. [51], 2012 Australia	RCT	Primary care	To test feasibility and impact of a multifactorial self-efficacy medication intervention to improve medication self-efficacy and adherence of culturally and linguistically diverse groups with DM, chronic kidney disease and chronic vascular diseases.	48 patients (EG: 26, CG: 22) Mean age (SD): 74.31 (8.37) M: 62.5% Europeans (Greek and Italian): 81.3%	a. Mean (SD): 3.29 (2.18); b. chronic kidney disease, DM and CVD	Self-efficacy and medication adherence	Primary: medication self-efficacy, medication adherence Secondary: general wellbeing, healthcare utilization and clinical laboratory, attrition rate	Self-efficacy medication intervention including individualized medication review, PowerPoint presentation, and MI phone contact. 3 months	No significant differences in medication self-efficacy, medication adherence, wellbeing, healthcare services use, and clinical laboratory tests were found. The attrition rate was high

ACEI: angiotensin-converting enzyme inhibitors; ARB: angiotensin II receptor blockers; CG control group; COPD: chronic obstructive pulmonary disease; CVD: cardiovascular diseases; DM: diabetes mellitus; EG: experimental group; HbA1c: glycated hemoglobin; HF: heart failure; IQR: interquartile range; M: male; MI: motivational interviewing; ns: not specified; RCT: randomized control trial; SD: standard deviation. Note: Primary care was defined as the first point of contact with health care (e.g., general practice, community pharmacy).

**Table 2 healthcare-11-01681-t002:** Characteristics of the qualitative articles included in the review.

Reference Country	Design	Setting	Aim	Participants	Chronic Conditions a. Number b. Type	Data Collection	Data Analysis	Findings
Brandberg et al. [54], 2021 Sweden	Longitudinal qualitative study	Primary care	To describe the process of and challenges to self-management activities as expressed by patients with multimorbidity in a 4-week post-discharge MI consultation trial.	16 participants Mean age (SD): 71 (10) M: 56.25%	a. Mean: 5 b. HF or COPD and at least one other chronic condition	Individual interview by phone or face-to-face	Inductive content analysis	Category 1 Managing system-centered care. Category 2 Handling the burden of living with multiple illnesses at home post-discharge.
Halloway et al. [60], 2020 USA	Mixed method: Qualitative descriptive	Outpatient clinic	To identify barriers to and motivators of participation in a lifestyle physical activity intervention, and obtain suggestions to make the program more appealing to older women with cardiovascular diseases.	18 female participants Mean age (SD): 73.9 (5.3) African American: (61.1%) Non-Hispanic white: (38.9%)	a. Mean (SD): 3.2 (0.9) b. Coronary heart disease, hypertension, atrial fibrillation	Focus group	Thematic analysis	Theme 1 Program expectations Theme 2 Barriers to lifestyle physical activity Theme 3 Motivators of lifestyle physical activity Theme 4 Strategies for increasing lifestyle physical activity Theme 5 Personal benefits of increasing lifestyle physical activity.
Williams and Manias [49], 2014 Australia	Qualitative	Primary care	To explore motivation and confidence of people with DM, chronic kidney disease and hypertension to take their medicines as prescribed.	39 participants Mean age (SD): 68 (8.3) M: 56.4% Non-Australian: 64.1%	a. Mean (SD): 7.7 (2.5) * b. DM, chronic kidney disease and hypertension	Individual interview by phone Transcribed verbatim handwritten notes taken during each call	Framework approach using the health belief model	Individual perception Theme 1 Importance of health: Wanting control of health; Alters medicine prescriptions; Use of complementary and alternative medicines. Theme 2 Perceived seriousness of disease: Thinking about mortality; Comorbidities complicate treatment; Acute illnesses risk health. Theme 3 Perceived threat of disease: Wishing and hoping; Denial—a cavalier approach; Looking to blame; Que sera sera. Modifying factors Theme 1 Demographic and psychosocial aspects: CALD influences; Family problems; Lack of resources. Theme 2 Interpersonal aspects: Partnerships with health professionals; Negotiating the healthcare maze; Difficulties with continuity of care. Theme 3 Cues to action: Learnt from prior experience; Seeking information; Self-efficacy; Positive feedback; Importance of a positive approach: mind over matter. Likelihood of action Theme 1 Perceived benefits of acting: Valuing medicines. Theme 2 Perceived barriers to act: Cost of medicines; Too many medicines; No symptoms; Medicine side effects; Forgetting medicines; Actively resisting medicines; Wary of changes to medicines; Uninformed about health matters; Questioning the benefit of medicines; Targets unrealistic. Theme 3 Likelihood of taking action: Having a supply of medicines; Support from family; Medicine reminders and routine
Willimas et al. [50], 2015 Australia	Qualitative	Primary care	To examine the perceptions of culturally and linguistically diverse participants with DM, chronic kidney disease and cardiovascular disease and to determine factors that influence medication self-efficacy using MI	26 participants Mean age (SD): 73.5 (9.14) M: 61.5% European: 84.6% (Greek, Italian), Vietnamese: 15.4%	a. Mean (SD): 3.5 (2.78) ***** b. DM, chronic kidney disease and CVD	Individual interview by phone Transcribed verbatim handwritten notes taken during each call	Framework approach	Theme 1 Attitudes toward medication. Subtheme 1 Appreciates medications. Subtheme 2 Burden of having to take medication. Subtheme 3 Distrust of medications. Subthemes 4 Concerned about side effects of medication. Theme 2 Having to take medication. Subtheme 5 Behavior that assists medication taking. Subtheme 6 Forgetting to take medication. Subtheme 7 Family support helpful. Subtheme 8 Hoping not to become worse. Theme 3 Impediment to chronic illness medications’ self-efficacy Subtheme 9 Insufficient knowledge of medication. Subtheme 10 Blind faith in medical advice. Subtheme 11 Medications are overwhelming. Subtheme 12 Cost of medication.

Legend: COPD: chronic obstructive pulmonary disease; CVD: cardiovascular diseases; HF: heart failure; DM: diabetes mellitus; GP: general practitioners; M: male; MI: motivational interviewing; SD: standard deviation. * in addition to the indexed diseases.

**Table 3 healthcare-11-01681-t003:** Obstacles and motivators to self-care behavior changes derived from qualitative studies.

Factors	Obstacles	Motivators
Healthcare system	Lack of continuity of care Healthcare services oriented to single chronic conditions Several healthcare providers in charge Not connected medical record systems Pharmaceutical policy	
Personal	Beliefs about health Perceptions about medication benefits Negative attitudes toward medications Personal and family experiences with medications Denial of health problems	Reaching autonomy Importance of health Positive attitude toward medications
Clinical	Complex treatments Number of medications Number of chronic conditions Co-occurrence of acute illnesses Difficulty to identify symptoms of each disease Absence of symptoms Side effects of medications Fear of dependency on medications Changes in prescriptions Forgetfulness Fatigue Impaired physical functions	Improving global health Not becoming sicker Staying alive Relief from symptoms Successful achievements of goals Personal reminder systems for medication Consolidated medication routine
Psychological	Lack of confidence Life restrictions	Self-efficacy Positive life approach
Relational	Lack of trust in healthcare providers	Trust in healthcare providers
Social	Cultural background Family obligations Lack of social support Low health literacy Social life restrictions caused by medications	Improved social life Support from family
Economic	Cost of treatment	Economic benefits

## Data Availability

The labelled dataset used to support the findings of this study is available from the corresponding author upon request.

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
