# Peer review of "Use of Motivational Interviewing in Older Patients with Multiple Chronic Conditions and Their Informal Caregivers: A Scoping Review"

_healthcare, 2023, doi:10.3390/healthcare11121681_

Round 1
Reviewer 1 Report
Dear Authors,
I am content to be a reviewer for the research entitled "Use of motivational interviewing in older adults with multiple chronic conditions and their informal caregivers: a scoping review."
The objective of this article is to clarify the extent and variety of evidence supporting the use of the motivational interview as a tool for improving the condition of elderly patients with multiple chronic conditions and to identify the gaps in knowledge in this area, which are also the purposes of systematic reviews as stated by the PRISMA-ScR. The abstract is well structured and provides the key findings and conclusions of the study, including the effectiveness of motivational interviewing in improving medication adherence with multiple chronic conditions, as well as the need for further research to explore its potential benefits in other areas, such as the potential contributor role of motivational interviewing on the part of caregivers.
The introduction provides the motivation to undertake the approach to the theme debated in this review, identifies the main inconsistencies from the previous studies, and states the main questions addressed by the authors.
The methodology gives extensive details and is adequate for the scope of the review and PRISMA recommendations. One of the exclusion criteria was studies on patients with dementia and psychiatric conditions (lines 99–100). Please state if you included studies on the MI caregivers of these patients, since it's not clearly stated. It would be helpful to include a short description of the stages of change according to trans-theoretical model of behavior change and its relation with MI in the concept section.
Line 234 – Provider training; line 241 - Training effectiveness assessment; Line 247 - Exposure time; Line 250 - treatment fidelity assessment These should be at the beginning of a new paragraph.
The clarity of Section 3.6 would be improved if the primary findings of the reviewed studies were presented according to the categories of Table 2, which are also listed at lines 271-272.
The discussion section identifies the primary limitations of the included studies as well as directions for future research.
In conclusion, I consider that the proposed scoping review entitled "Use of motivational interviewing in older adults with multiple chronic conditions and their informal caregivers: a scoping review" should be reconsider after major revision.
Sincerely yours,
Author Response
Dear Editor and Reviewers,
Thank you for the opportunity to submit a revised draft of our manuscript. We greatly appreciate the time and effort you dedicated to providing your feedback on our manuscript. We have incorporated the changes to reflect most of the suggestions. You can find below the detailed responses to your comments, and the changes directly in the paper highlighted in red.
|
Reviewer 1 Dear Authors, I am content to be a reviewer for the research entitled "Use of motivational interviewing in older adults with multiple chronic conditions and their informal caregivers: a scoping review." The objective of this article is to clarify the extent and variety of evidence supporting the use of the motivational interview as a tool for improving the condition of elderly patients with multiple chronic conditions and to identify the gaps in knowledge in this area, which are also the purposes of systematic reviews as stated by the PRISMA-ScR. The abstract is well structured and provides the key findings and conclusions of the study, including the effectiveness of motivational interviewing in improving medication adherence with multiple chronic conditions, as well as the need for further research to explore its potential benefits in other areas, such as the potential contributor role of motivational interviewing on the part of caregivers. The introduction provides the motivation to undertake the approach to the theme debated in this review, identifies the main inconsistencies from the previous studies, and states the main questions addressed by the authors. The methodology gives extensive details and is adequate for the scope of the review and PRISMA recommendations
|
Thanks for your comments, which have greatly contributed to the improvement of the paper. See below the manuscript revised according to your suggestions.
|
|
One of the exclusion criteria was studies on patients with dementia and psychiatric conditions (lines 99–100). Please state if you included studies on the MI caregivers of these patients, since it's not clearly stated.
|
Thank you for the suggestion. We specified that we excluded caregivers of patients with dementia and psychiatric disorders, as well as caregivers with such conditions, to explain the reasons.
|
|
Results It would be helpful to include a short description of the stages of change according to trans-theoretical model of behavior change and its relation with MI in the concept section.
|
We shortly introduced the stage of change, as you suggested, in the context of the use of MI within theoretical frameworks.
|
|
Line 234 – Provider training; line 241 - Training effectiveness assessment; Line 247 - Exposure time; Line 250 - treatment fidelity assessment These should be at the beginning of a new paragraph.
|
Thanks for your suggestion, which we have followed. Moreover, we have highlighted the title of subsections in italics to make them more readable.
|
|
The clarity of Section 3.6 would be improved if the primary findings of the reviewed studies were presented according to the categories of Table 2, which are also listed at lines 271-272.
|
We presented the qualitative results according to the identified factors, as suggested. This helped shorten the results section and make them more appropriate for the aim of the paper. We also have changed the name of the paragraph, as suggested by another reviewer (Factors influencing motivation to change self-care behaviors)
|
|
The discussion section identifies the primary limitations of the included studies as well as directions for future research. In conclusion, I consider that the proposed scoping review entitled "Use of motivational interviewing in older adults with multiple chronic conditions and their informal caregivers: a scoping review" should be reconsidered after major revision. |
|
Reviewer 2 Report
I thank the authors of this study for their work and this study. This study, a scoping review, explores the impact of using motivational interviewing (MI) with older adults living with multiple chronic conditions (MCC), and their informal caregivers. According to the authors, this is a known but new intervention, in terms of it being used and implemented with this patient population. They consider it a relevant and impactful intervention to achieve health-related behaviors changes and therefore, its high relevance for clinical practice. This scoping review appears to be the first step of a larger research program, where it was first proposed that the state of the literature, science and knowledge, on uses of this intervention with this participant groups be reviewed and assessed.
In terms of positive points: the paper is well written. Please reread throughout for typos, but in general, it is very well written and clear. However, I would recommend creating a few more paragraph breaks. In the results section in particular, some sections are hard to read. I would recommend, for the quantitative studies, that you either use points forms OR bold/underline texts, to the points you are covered; in the qualitative studies section, a few more paragraph breaks.
The methods and methodology are well presented and clear. The study design is pertinent, as well as the research question, well formulated and relevant. The study design is well understood and mastered.
In your methods, however, please define better 'self-care behaviors'. I do come back to this point in my criticisms/review suggestions.
I have a number of comments regarding this paper. I do consider, however, and recommend its publication, but with some revisions I consider minor.
First, smaller points
· The authors indicated that the time frame for their inclusion/exclusion of published studies was the past ten years 2012-2022. Two questions: First, why?; second, in the final parts of the paper, the authors indicated that this is a relatively new intervention, as the studies date in the past ten years. This is likely a wording issue – did you or not consider a larger timeframe? If so, please indicate clearly. If not, please justify and clarify your conclusions, given that this is not a reflection of the science, but your method.
· Introduction and justification: I consider that more references are needed regarding MI, as an intervention, with ppl with chronic illness, self-management of health, use of medicine, pain; and caregivers. There are currently only a handful of studies presented, regarding what MI is, how and why it emerged, in which contexts (clinicals); and an idea of the scope of the literature/application of this intervention in healthcare, even if only rough. Otherwise, the introduction gives us the sense that this is a rarely used intervention, and in very specific contexts.
· Justification of MI: I struggle with the insistence of MI as a relevant and ‘good’ intervention, especially given the little that is said/presented in this study regarding this intervention. This ‘ambivalence’ I felt only grew strongly as well as I reached the findings and specifically, the discussion part. I got a sense that MI lends itself to critique, specifically that it could be seen as an intervention aiming to increase patient/users compliance, and therefore, is not really person-centre/family centred. In the discussion, the authors actually also themselves make that claim, when they indicate that MI is mainly used (and useful) to get patients and families to adhere to medication. It seems to me, therefore, that there are many criticisms that can be addressed to this intervention, and I recommend that the authors spend a few paragraphs on this, i.e. presenting in more details MI, applications, how it is or not personcentred; and the criticisms from the literature; and why they do consider it relevant and useful.
· In relation to this point, I also would recommend that the authors define more strongly what they consider ‘health-related behaviors changes”, because taking/adherence to prescribed medicine, in a context where we see a lot more of discussion/push toward demedicalisation and deprescription, and non-pharmacology approaches, it seems MI is not really ‘diverse’ in terms of the ‘health-realted behaviors’ it speaks of. Secondly, I am curious ‘whose’ version of ‘health-related behaviors’ are proposed with MI – health professionals or patients and their families? In other words – how is MI person-centred? Or simply a method to achieve ‘patient compliance and adherence’ to a mode of treatment that is imposed by health professionals? Put yet otherwise – is MI a two-way conversation?
· Inclusion and exclusion: I struggle quite strongly with your decisions, without in my view adequate justification, for excluding patients with neurocognitive decline and psychiatric conditions, especially if you want to include informal caregivers. You seem also to operate on problematic, preconceived and inappropriate views of older adults living with neurocognitive degenerative conditions and psychiatric conditions. Please refer to following works for an Enlighted views of people living with neurodegenerative and cognitive conditions:
o Ward, R., & Sandberg, L. J. (Eds.). (2023). Critical dementia studies : an introduction (Ser. Dementia in critical dialogue). Routledge.
o Bartlett, R., & O’Connor, D. (2010). Broadening the Dementia Debate: Towards Social Citizenship. Policy Press.
o Leibing, A., & Cohen, L. (2006). Thinking About Dementia: Culture, Loss, and the Anthropology of Senility. Rutgers University Press. - idem for psychiatric conditions
· Context. In your inclusion criteria – authors indicate that studies conducted in any healthcare setting (i.e., hospital, long-term care) and primary care and community, and in any country and culture were included. – Please indicated contexts that were excluded. I would think social services and community services could have been a place to yield relevant findings.
· Exclusion. I do not understand your decision to “exclude(d) systematic, narrative, or scoping reviews” even though you “screened their reference lists to identify additional pertinent articles." – plus, it would be good to explain how many you may have identified (of these reviews).
· Further details on how, why and by whom MI is implemented would be useful
· Use of ‘could’ confusion, in presentation of qualitative studies. Please reread carefully. When reading your use of ‘could’, with the qualitative findings, I was often confused/unsure if these were your views, on the findings, or what was in the studies’ findings (that you are reviewing). In any case, the use of ‘could’ is unclear – expressed by participants? Etc.
· Qualitative assessments. When reading these findings, I find myself unclear and asking myself questions regarding the objectives of your studies. It seems the findings here are not about MI, or its uses, or its impacts, but rather on patients’ viewpoint – which is greatly relevant, and to your work! But I think it needs some rewording, i.e.: ‘types of qualitative findings when MI is used’, or something like that; rather than ‘impact of using MI’, because it is not so much an ‘impact’ (i.e. increasing health-related behavior changes”, but rather contextual information about patients, their experience and background, which is relevant information to ‘adjust’ an intervention or healthplan
o I also find quite ‘thin’ the findings emerging from the qualitative studies – is it just qualitative data that is presented? What findings regarding the uses of MI emerged from the qualitative studies? There are different ‘levels’ of findings, and as it stands, the presentations of the findings isn’t clear.
· Clarifying findings. Ultimately, this is a larger issue/criticism I have, i.e., that there are confusiong between the following: (1.1) documenting the impact MI as intervention has in term of increasing health-related behaviors; (1.2) its impact with regard to this objective compared to other methods or intervention; OR (2) what impact the uses of MI has, including the types of data it can yield (from patients and informal carers); including: how it is used, when, where; what we learn from the findings (i.e. is it a ‘good’ intervention? worth using? Which contexts? should it be complementary? To what?) etc.
· Combinaison? In this sense, it would be worthwhile to discuss what other interventions were used, (ie. combinaison). More data on this would be helpful
· Discussion. I find the discussion too repetitive of the findings, and not engaging enough with the existing literature; also – what further recommendations re: MI do you make, notably about its actually content?
· Negotiating your personal investment in the intervention? I would like more critical engagement with MI. At this time, as written, the authors appear ‘too convinced’ yet with unsufficient data. Why do you recommend it? For what concrete health-related changes? And is adherence to medicine always a ‘health-related positive behavior’, especially given what emerged from the qualitative studies you reported on.
· Line 398 - 'it was used with patients who were very old' -- what does that mean or imply?
· Line 403-404 -- important findings re: what can or not be attributed to MI...
o who should used and implemented? providers?
· What do patients and families think of the intervention itself?
· Line 462 - could you expand? Re: included studies, that contributed little (to review)-- why?
· Conclusion: Authors say that MI could contribute greatly to health-related behaviors changes -- but without enough studies, can this be asserted?, especially as you also indicate that it is unclear, in your included studies, what can fully be attributed to MI – how can you assert this? Based on what?
· Supplement 1 -- needed?
· Supplement 4 - include, if possible, nb of participants, patients/family members/informal caregivers AND carers/health professionals/providers (if later is relevant)
Thank you! I hope these comments are helpful. I am available for reviewing a revised manuscript.
None, see above
Author Response
Dear Editor and Reviewers,
Thank you for the opportunity to submit a revised draft of our manuscript. We greatly appreciate the time and effort you dedicated to providing your feedback on our manuscript. We have incorporated the changes to reflect most of the suggestions. You can find below the detailed responses to your comments, and the changes directly in the paper highlighted in red.
|
Reviewer 2 I thank the authors of this study for their work and this study. This study, a scoping review, explores the impact of using motivational interviewing (MI) with older adults living with multiple chronic conditions (MCC), and their informal caregivers. According to the authors, this is a known but new intervention, in terms of it being used and implemented with this patient population. They consider it a relevant and impactful intervention to achieve health-related behaviors changes and therefore, its high relevance for clinical practice. This scoping review appears to be the first step of a larger research program, where it was first proposed that the state of the literature, science and knowledge, on uses of this intervention with this participant groups be reviewed and assessed. In terms of positive points: the paper is well written.
|
Thank you for your favorable remarks. We followed your suggestions through the text.
|
|
Please reread throughout for typos, but in general, it is very well written and clear. However, I would recommend creating a few more paragraph breaks. In the results section in particular, some sections are hard to read. I would recommend, for the quantitative studies, that you either use points forms OR bold/underline texts, to the points you are covered; in the qualitative studies section, a few more paragraph breaks
|
We checked for typos as you suggested. We have broken the text in more paragraphs in the results and discussion sections, as suggested.
|
|
The methods and methodology are well presented and clear. The study design is pertinent, as well as the research question, well formulated and relevant. The study design is well understood and mastered. In your methods, however, please define better 'self-care behaviors'. I do come back to this point in my criticisms/review suggestions. I have a number of comments regarding this paper. I do consider, however, and recommend its publication, but with some revisions I consider minor.
|
We have provided a more complete self-care behavior definition in the introduction section. When we first introduced the terms “self-care behaviors”, according to Middle-Range Theory of Self-Care in Chronic Illness |
|
First, smaller points The authors indicated that the time frame for their inclusion/exclusion of published studies was the past ten years 2012-2022. Two questions: First, why?; second, in the final parts of the paper, the authors indicated that this is a relatively new intervention, as the studies date in the past ten years. This is likely a wording issue – did you or not consider a larger timeframe? If so, please indicate clearly. If not, please justify and clarify your conclusions, given that this is not a reflection of the science, but your method.
|
Actually, we stated that in our search, we included studies from inception to July 2022. Probably you missed this information. We have highlighted this sentence in red, and you can find it at line 192. Instead, the articles we found were published from 2012-2022.
|
|
Introduction and justification: I consider that more references are needed regarding MI, as an intervention, with ppl with chronic illness, self-management of health, use of medicine, pain; and caregivers. There are currently only a handful of studies presented, regarding what MI is, how and why it emerged, in which contexts (clinicals); and an idea of the scope of the literature/application of this intervention in healthcare, even if only rough. Otherwise, the introduction gives us the sense that this is a rarely used intervention, and in very specific contexts.
|
Thank you for your suggestion. We have added more literature on the use of MI with patients and caregivers in many conditions to illustrate its extensive use.
|
|
Justification of MI: I struggle with the insistence of MI as a relevant and ‘good’ intervention, especially given the little that is said/presented in this study regarding this intervention. This ‘ambivalence’ I felt only grew strongly as well as I reached the findings and specifically, the discussion part. I got a sense that MI lends itself to critique, specifically that it could be seen as an intervention aiming to increase patient/users compliance, and therefore, is not really person-centre/family centred. In the discussion, the authors actually also themselves make that claim, when they indicate that MI is mainly used (and useful) to get patients and families to adhere to medication. It seems to me, therefore, that there are many criticisms that can be addressed to this intervention, and I recommend that the authors spend a few paragraphs on this, i.e. presenting in more details MI applications, how it is or not person centred; and the criticisms from the literature; and why they do consider it relevant and useful.
|
We tried to solve the ambivalence you perceived reading the paper. In the introduction, we stated that it is a person-centered intervention that has had a broad use in health, with promising results in many conditions, and in the discussion we present the limits in its use. Although MI can be used to support behavior changes in many conditions, our review finds its use has been limited to modify behaviors deemed most important by healthcare professionals, such as medication adherence, with little attention paid to improve other self-care behaviors that may be relevant to patients. We hope to have clarified better this point. |
|
In relation to this point, I also would recommend that the authors define more strongly what they consider ‘health-related behaviors changes”, because taking/adherence to prescribed medicine, in a context where we see a lot more of discussion/push toward demedicalisation and deprescription, and non-pharmacology approaches, it seems MI is not really ‘diverse’ in terms of the ‘health-related behaviors’ it speaks of. Secondly, I am curious ‘whose’ version of ‘health-related behaviors’ are proposed with MI – health professionals or patients and their families? In other words – how is MI person-centred? Or simply a method to achieve ‘patient compliance and adherence’ to a mode of treatment that is imposed by health professionals? Put yet otherwise – is MI a two-way conversation?
|
In the concept section of the “Eligibility Criteria”, we defined what we meant with self-care behaviors changes and by whom these changes can be driven (lines 145-151). This concept was also described in the discussion section. We hope to have clarified this critical aspect. To be consistent, we replaced the expression “health-related behaviors” with “self-care behaviors”, which are the behaviors that we have addressed in our review.
|
|
Inclusion and exclusion: I struggle quite strongly with your decisions, without in my view adequate justification, for excluding patients with neurocognitive decline and psychiatric conditions, especially if you want to include informal caregivers. You seem also to operate on problematic, preconceived and inappropriate views of older adults living with neurocognitive degenerative conditions and psychiatric conditions. Please refer to following works for an Enlighted views of people living with neurodegenerative and cognitive conditions: o Ward, R., & Sandberg, L. J. (Eds.). (2023). Critical dementia studies: an introduction (Ser. Dementia in critical dialogue). Routledge. o Bartlett, R., & O’Connor, D. (2010). Broadening the Dementia Debate: Towards Social Citizenship. Policy Press. Leibing, A., & Cohen, L. (2006). Thinking About Dementia: Culture, Loss, and the Anthropology of Senility. Rutgers University Press. - idem for psychiatric conditions |
We think you have highlighted an interesting issue. To date, MI has not been used with patients with a diagnosis of dementia and major psychiatric conditions, and with their caregivers, as these conditions can alter awareness of health problems, impair patient reasoning and problem-solved skills, making it difficult to conduct MI intervention. These conditions are usually criteria for exclusion from the studies (see https://www.jmcp.org/doi/full/10.18553/jmcp.2017.23.5.549). However, a few studies have been conducted on older people with mild or moderate cognitive impairment or at risk of dementia, with encouraging results (see for example ohttps://www.sciencedirect.com/science/article/pii/S235287371930068X). However, these studies were not included in the review because they did not consider or describe the presence of other chronic conditions, which was an inclusion criterion for us. Please, if you know studies that have used MI in patients with dementia and other chronic conditions, let us know that we can update our review. |
|
Context. In your inclusion criteria – authors indicate that studies conducted in any healthcare setting (i.e., hospital, long-term care) and primary care and community, and in any country and culture were included. – Please indicated contexts that were excluded. I would think social services and community services could have been a place to yield relevant findings. |
No context was excluded. We reworded the sentence to make clearer this point.
|
|
Exclusion. I do not understand your decision to “exclude(d) systematic, narrative, or scoping reviews” even though you “screened their reference lists to identify additional pertinent articles." – plus, it would be good to explain how many you may have identified (of these reviews).
|
We included only primary studies related to the use of MI as they were those from which we could derive first-hand information, since in reviews primary studies are synthesized, not allowing extracting the data we are looking for. We listed the reviews we identified on MI in the introduction section where we reported the literature on the topic (from lines 82 to 92), describing their aims and results to show the lack of knowledge on this topic. Moreover, the JBI methodology recommends to screen the reference list of all included sources to have a most comprehensive search (https://jbi.global/sites/default/files/2022-02/JBI_Protocol_Template_Scoping_Reviews.docx https://jbi.global/scoping-review-network/resources)
|
|
Further details on how, why and by whom MI is implemented would be useful
|
We highlighted such details in the sections 3.4 Targeted health-related behaviors and outcomes, and 3.5 Characteristics of the motivational interviewing interventions. We described these contents in separate subsections and paragraphs to make clearer the contents describing MI providers, how and aims.
|
|
·Use of ‘could’ confusion, in presentation of qualitative studies. Please reread carefully. When reading your use of ‘could’, with the qualitative findings, I was often confused/unsure if these were your views, on the findings, or what was in the studies’ findings (that you are reviewing). In any case, the use of ‘could’ is unclear – expressed by participants? Etc.
|
We agree with your point that using “could” could generate confusion. We reworded the presentation of qualitative results, and, following the suggestion of another reviewer, we reported the results according to the factors instead of the authors, eliminating the use of “could”.
|
|
Qualitative assessments. When reading these findings, I find myself unclear and asking myself questions regarding the objectives of your studies. It seems the findings here are not about MI, or its uses, or its impacts, but rather on patients’ viewpoint – which is greatly relevant, and to your work! But I think it needs some rewording, i.e.: ‘types of qualitative findings when MI is used’, or something like that; rather than ‘impact of using MI’, because it is not so much an ‘impact’ (i.e. increasing health-related behavior changes”, but rather contextual information about patients, their experience and background, which is relevant information to ‘adjust’ an intervention or health plan I also find quite ‘thin’ the findings emerging from the qualitative studies – is it just qualitative data that is presented? What findings regarding the uses of MI emerged from the qualitative studies? There are different ‘levels’ of findings, and as it stands, the presentations of the findings isn’t clear.
|
We changed the title of paragraph title in “factors influencing the motivation to change self-care behavior” and, as suggested by another review, we presented the results according to such factors. We clarified the scope of including qualitative studies, also adding it in the description of the aim of the study.
|
|
·Clarifying findings. Ultimately, this is a larger issue/criticism I have, i.e., that there are confusion between the following: (1.1) documenting the impact MI as intervention has in term of increasing health-related behaviors; (1.2) its impact with regard to this objective compared to other methods or intervention; OR (2) what impact the uses of MI has, including the types of data it can yield (from patients and informal carers); including: how it is used, when, where; what we learn from the findings (i.e. is it a ‘good’ intervention? worth using? Which contexts? should it be complementary? To what?) etc. |
The scope of our review, consistent with the scoping review method, was to map the literature on the topic, and we reported how it was used (in which chronic conditions, with which patients, for which self-care behaviors, etc.). We did not want to take any position regarding its impact or effectiveness, but we wanted to identify possible lack in the literature. We made some changes to the discussion session to clear our scope to the readers. |
|
·Combinaison? In this sense, it would be worthwhile to discuss what other interventions were used, (ie. combinaison). More data on this would be helpful
|
We do not know if you are asking to describe other interventions used to modify self-care behaviors. In that case, it would be the aim of another review in which more interventions are compared. If you meant to add more details about the other interventions with which it was combined, we briefly discussed this from line 474-484
|
|
·Discussion. I find the discussion too repetitive of the findings, and not engaging enough with the existing literature; also – what further recommendations re: MI do you make, notably about its actually content?
|
We made some changes at the discussion section, eliminating repetitions and made clearer our recommendations.
|
|
Negotiating your personal investment in the intervention? I would like more critical engagement with MI. At this time, as written, the authors appear ‘too convinced’ yet with unsufficient data. Why do you recommend it? For what concrete health-related changes? And is adherence to medicine always a ‘health-related positive behavior’, especially given what emerged from the qualitative studies you reported on.
|
Actually, we did not want to recommend using this method, as we do not have enough data to do this, and it was not the scope of our scoping review. We suggested further studies on the critical topics we identified. The changes we made in discussion and conclusion should have clarified our position.
|
|
·Line 398 - 'it was used with patients who were very old' -- what does that mean or imply?
|
We wanted to highlight that MI can be used to motivate patients at any age, as the method was applied to very old patients.
|
|
·Line 403-404 -- important findings re: what can or not be attributed to MI... o who should used and implemented? providers? What do patients and families think of the intervention itself?
|
We added further details about these aspects, as you suggested.
|
|
Line 462 - could you expand? Re: included studies, that contributed little (to review)-- why?
|
We specified why they contributed little to the scope of the review, due to the scarce information provided on the application of the method.
|
|
Conclusion: Authors say that MI could contribute greatly to health-related behaviors changes -- but without enough studies, can this be asserted?, especially as you also indicate that it is unclear, in your included studies, what can fully be attributed to MI – how can you assert this? Based on what?
|
We reworded the conclusion section to illustrate better what we found with our scoping review, as previously described.
|
|
Supplement 1 -- needed?
|
Nowadays, the reporting check lists are strongly recommended or considered mandatory by many journals in the submission process to testify the quality of the study reporting. We decided to adhere at such recommendations, and we included the checklist in the submission for transparency.
|
|
Supplement 4 - include, if possible, nb of participants, patients/family members/informal caregivers AND carers/health professionals/providers (if later is relevant) |
Such information was already presented in Table 1 and Table 2 and we did not repeat them in supplementary Table 4.
|
|
Thank you! I hope these comments are helpful. I am available for reviewing a revised manuscript. |
Thanks! |
Reviewer 3 Report
This is an important study summarizing motivational interviewing used to change health behaviors in older adults with multiple chronic illnesses. I believe it adds to the literature and is an important review article. However, there are some minor additions that should be made.
Overall:
Minor English language checks should be conducted.
Introduction:
Line 38 – Rewrite sentence – it is confusing as it currently leads the reader to believe that the informal caregiver replaces the patient partially or completely to manage their disease.
What is known about MI use in older patients in general? What is known about MI use for caregivers of older patients with any number of chronic conditions? Authors should include a summary of literature focusing on older adults, not necessarily those with multiple conditions, but research focused on MI among older adults and informal caregivers of older adults.
Materials and Methods:
What is JBI methodology? What does JBI stand for?
Authors should provide more justification for their methods, namely: 1) why include studies where more than 80% of the enrolled patients were aged 65 and older; 2) why was the list of chronic conditions developed by OASH used; 3) why exclude studies that considered motivation as an outcome of the intervention?
Was the search limited by publication date? In the abstract, authors mentioned that studies conducted between 2012 and 2022 were included in the study.
Results:
The abstract states that 12 studies were identified, but the Results sections states that 15 articles were identified. Can authors clarify which is correct?
Discussion:
Authors should break up the discussion sections into smaller paragraphs.
How can you justify the conclusion that informal caregivers should be targeted in MI interventions as they are essential for the care of older patients with MCCs (conclusion included in the abstract as well)? Authors have not provided any information on the importance of informal caregivers in changing health behaviors for older adults, even for those with only one chronic condition. Authors should provide additional justification for this conclusion.
Very minor grammatical issues should be addressed.
Author Response
Dear Editor and Reviewers,
Thank you for the opportunity to submit a revised draft of our manuscript. We greatly appreciate the time and effort you dedicated to providing your feedback on our manuscript. We have incorporated the changes to reflect most of the suggestions. You can find below the detailed responses to your comments, and the changes directly in the paper highlighted in red.
|
Reviewer 3 Comments and Suggestions for Authors. This is an important study summarizing motivational interviewing used to change health behaviors in older adults with multiple chronic illnesses. I believe it adds to the literature and is an important review article. However, there are some minor additions that should be made. Overall: Minor English language checks should be conducted.
|
Thanks for your positive comments. We checked the text for English errors.
|
|
Introduction: Line 38 – Rewrite sentence – it is confusing as it currently leads the reader to believe that the informal caregiver replaces the patient partially or completely to manage their disease.
|
Actually, it was what we meant, that they can replace the patients in self-care behaviors when needed. We explained better the caregiver contribution to patient self-care in the discussion section: when patients could not perform self-care by themselves, informal caregivers replace them in managing the disease (for example, administering medications, controlling blood pressure or glycemia, calling the physicians in case of exacerbations).
|
|
What is known about MI use in older patients in general? What is known about MI use for caregivers of older patients with any number of chronic conditions? Authors should include a summary of literature focusing on older adults, not necessarily those with multiple conditions, but research focused on MI among older adults and informal caregivers of older adults.
|
We have broadened the literature on the use of MI in other conditions, as suggested by you and another reviewer.
|
|
Materials and Methods: What is JBI methodology? What does JBI stand for?
|
JBI was formerly known as the Joanna Briggs Institute, but nowadays they use the name JBI (no more considered an acronym), as it is no more affiliated to this institute. We specified this in the text for clarity. |
|
Authors should provide more justification for their methods, namely: 1) why include studies where more than 80% of the enrolled patients were aged 65 and older; 2) why was the list of chronic conditions developed by OASH used; 3) why exclude studies that considered motivation as an outcome of the intervention?
|
1) We have specified why we decided to include studies with more than 80% patients aged 65 and over at line 119.121. 2) We specified the reason for our choice of using OSAH as a classification system for chronic diseases 3) We have specified the reason for not considering studies on motivation at line 161-163.
|
|
Was the search limited by publication date? In the abstract, authors mentioned that studies conducted between 2012 and 2022 were included in the study. |
No limitation to publication date was set. We specified in the manuscript that the search was conducted from inception to July 2022. We have also reported this information in the abstract for clarity. The 2012-2022 was the date range of articles that we found.
|
|
Results: The abstract states that 12 studies were identified, but the Results sections states that 15 articles were identified. Can authors clarify which is correct? |
We distinguish between published articles and studies from which the articles were derived. For clarity and consistency, we have also specified in the abstract that we found 15 articles reporting 12 studies. We explained what we meant in the results section: “Two articles presented the results of a study conducted on the same sample describing two different outcomes, and two qualitative articles presented an analysis of the contents of the MI sessions of two quantitative studies”.
|
|
Discussion: Authors should break up the discussion sections into smaller paragraphs.
|
Thank you for your suggestion that we have followed.
|
|
How can you justify the conclusion that informal caregivers should be targeted in MI interventions as they are essential for the care of older patients with MCCs (conclusion included in the abstract as well)? Authors have not provided any information on the importance of informal caregivers in changing health behaviors for older adults, even for those with only one chronic condition. Authors should provide additional justification for this conclusion. |
As suggested, we have provided more information on the role of informal caregiver in the discussion section to justify our conclusion (line 513-528). |
Round 2
Reviewer 1 Report
To the Authors,
Thank you for the opportunity to revise the manuscript entitled "Use of motivational interviewing in older adults with multiple chronic conditions and their informal caregivers: a scoping review" which addresses the role of the motivational interview in improving the self-care of old patients with multiple comorbidities. The manuscript is well-structured and provides a rigorous methodology. The results and the discussions are well-organized and address the description of the characteristics of the motivational interviewing interventions included in the review and the factors influencing motivation to change self-care behaviors. Moreover, the limitations of current studies (few instruments that measure interviewers’ adherence, lack of regarding the training process of MI providers, and the necessity to adhere to a standardized checklist to guide the MI) are presented, indicating a gap in this field.
Therefore I consider the manuscript entitled "Use of motivational interviewing in older adults with multiple chronic conditions and their informal caregivers: a scoping review" suitable to be published in the Healthcare Journal.
Sincerely
Author Response
Thanks. Your suggestions were very helpful.
Reviewer 3 Report
The authors have adequately addressed my concerns.
Author Response
Thanks. Your suggestions were very helpful